# Home Range, Movement, and Nest Use of Hedgehogs (*Erinaceus europaeus)* in an Urban Environment Prior to Hibernation

**DOI:** 10.3390/ani14010130

**Published:** 2023-12-29

**Authors:** Lars Mørch Korslund, Marius Stener Floden, Milla Mona Sophie Albertsen, Amalie Landsverk, Karen Margrete Vestgård Løkken, Beate Strøm Johansen

**Affiliations:** 1Department of Natural Sciences, University of Agder, Universitetsveien 25, 4630 Kristiansand, Norway; 2Natural History Museum and Botanical Garden, Department of Natural Sciences, University of Agder, Gimleveien 27, Gimle Gård, 4630 Kristiansand, Norway

**Keywords:** hedgehog conservation, human–wildlife, urban wildlife, radio telemetry, home range, movements, nest use, hibernation nest

## Abstract

**Simple Summary:**

Populations of West European hedgehogs (*Erinaceus europaeus*) are decreasing all over Europe, and we are urgently in need of more knowledge to understand the challenges they face. In the Nordic countries, the winter nest locations are of crucial importance for hedgehogs to survive the winter hibernation period. Using radio transmitters, we studied 9 adult hedgehogs during the pre-hibernation period from August–November in a typical residential area in the city of Kristiansand, Southern Norway. The hedgehogs had a highly variable home range size and displayed a large variation in distance moved per hour, with no clear difference between sexes. There were also large individual differences in the number of nest sites used and how often they changed nests. Although hedgehogs had nesting places in a variety of gardens and in hedgerows along roads, such places seemed to lack appropriate nesting materials, suggesting that this is not a habitat suitable for winter hibernation. In September, as they prepared for hibernation, hedgehogs rather chose permanent winter nests in natural forest patches within residential areas, often under tree roots. Our research highlights the importance of maintaining and increasing the number of smaller forested patches within urban regions to help conserve hedgehog populations.

**Abstract:**

The West European hedgehog *(Erinaceus europaeus)* is in decline, and it is important to identify its challenges. We used VHF-telemetry to monitor pre-hibernation space use, nest use, and hibernation sites in a suburban area in Norway. Based on nine adult hedgehogs tracked between August and November 2002, we found that home range size was not dependent on individual sex or weight and that home ranges overlapped between individuals regardless of sex. The distance moved was not dependent on individual sex, but there was a tendency for increased movement before dawn. The number of nests used per individual (0–10) and the number of nest switches (0–14) varied greatly and did not differ significantly between sexes. Out of 28 nest sites, 16 were linked to buildings and 12 to vegetation, and nesting material was most often grass and leaves. Three hedgehogs monitored until hibernation established winter nests under tree roots in natural forest patches in September, and this suggests that establishing or maintaining forest patches in urban areas is important to ensure suitable hibernation habitat for hedgehogs. Our study was limited by a low sample size, and additional research is required to gain a deeper understanding of the challenges hedgehogs face in urban environments.

## 1. Introduction

Urbanization is often referred to as the process of the increasing concentration of people in cities and the transformation of natural environments into urban areas [1]. Urban environments have been rapidly expanding globally as a result of high population growth over the last decades [2]. The effect of urbanization can be complex and varied, but it generally tends to have a negative impact on biodiversity both locally and globally, mainly through habitat loss and fragmentation [3]. Several ecological studies on urban systems show that urban centers often have low biodiversity, with a few resilient species in high numbers [4]. Synanthropic species, species of wild animals that live in close proximity to humans and in environments that humans create [5], often reach higher densities in urban environments than in the wild [6]. A higher abundance of food, a lower abundance of predators, or a combination of these can result in increased population densities [7]. Small to medium-sized animals seem best suited for urban environments [8], and one such mammal that is common in urban environments in Europe is the West European hedgehog (*Erinaceus europaeus*, from here on called hedgehog) [9].

Results from multiple monitoring programs show that hedgehog populations are declining in many European countries [10,11,12,13,14,15,16,17]. According to the IUCN Red List assessment, the hedgehog’s conservation status varies from “Near Threatened” in Sweden and Norway to “Vulnerable” in the UK and “Endangered” in the Netherlands [18,19,20]. It has previously been suggested that hedgehogs prefer to live in rural areas, but some studies report a substantial population decline in such areas and that hedgehogs prefer residential areas [7,15,16,21,22]. This decline in rural areas appears to be primarily caused by intensive agriculture and intraguild predators [16,23]. Hubert, Julliard, Biagianti, and Poulle [7] found that factors such as access to anthropogenic food sources and favorable micro-climatic conditions may be key indicators of the high hedgehog presence in urban areas, and garbage and food put out for pets or other animals are often available food sources for hedgehogs [9]. Hedgehogs living in urban environments tend to become active post-midnight and avoid foraging near roads as a response to human-related dangers such as pedestrians and vehicle traffic [24], and they are usually found in greenspaces such as parks, road verges, and gardens [25]. These habitats are well suited for hedgehogs, but the fragmentation between such habitats, caused by roads and fences, can pose a significant challenge to the survival of these populations [26]. The most important habitat in urban areas is private gardens [27], as these have a high structural complexity with different vegetation such as lawn, flowerbeds, hedges, and terraces, creating habitats for both nesting and foraging [9]. On and around gardens and lawns, hedgehogs can find valuable food such as insects, slugs, and snails [28]. However, gardens may pose a variety of threats to individual hedgehogs as well. Use of garden pesticides such as insecticides, molluscicides, and rodenticides will lead to a decrease in the availability of natural food sources and can also result in secondary poising [24]. Although hedgehogs are capable of swimming, garden ponds can pose a threat to them as they can drown if unable to find a way out onto solid ground [29,30], and uncovered window wells, basement stairs, tennis nets, and nets covering berry bushes can function as traps [14,31].

Hedgehogs are solitary mammals and are mostly active during the night, spending much of the day sleeping in a nest [32]. They are not territorial animals, and the home range of individuals of the same and opposite sexes can overlap (see, e.g., [32,33,34,35,36,37,38,39]). The size of the home range is dependent on food availability, season, and sex, and while some hedgehogs stay in the same area over several years, others may wander more erratically around [29]. Hedgehogs that live in less productive environments will typically have a larger home range in order to find enough food [38,40]. During spring, in the mating season, males can have considerably larger home ranges than females, while in autumn the differences get smaller, and females can even have larger home ranges than males [26,38]. This seasonal change in home range size is reflected in the travel distance, and while male hedgehogs are found to travel considerably longer distances than female hedgehogs during spring, distances are more similar in the post-mating season [36]. Doncaster, Rondinini, and Johnson [21] found that some hedgehogs can move up to 9.9 km in total during the night, but during the summertime, hedgehogs often move only 2–3 km a night to forage and build up fat storage for their hibernation through the winter [29].

Nests are important for the nocturnal and solitary hedgehog, both for hibernation, protection, and breeding [41], and nests can be divided into three categories: breeding nests for the females and their litter, day or summer nests that are used as shelter during the days in the active season, and hibernation or winter nests where they spend up to several months undergoing hibernation [32]. Winter nests in mixed woodland habitat in England were often located at sites with structural support, such as under bramble bush or piles of logs, and the nesting material often consisted of grass or leaves packed together up to 20 cm in thickness [42,43]. Rautio et al. [41] found that urban hedgehogs in Finland moved to pine woods to establish winter nests and hibernate under the roots of large pine trees. The hedgehog hibernation period varies greatly with the local climate and, thus, geography. In Southern Europe, it only lasts for about two months, from January to February [33], while in Fennoscandia, at the northern boundary of the geographical range, it can last more than 200 days, starting as early as mid-September [41,44,45,46].

In coastal Southern Norway, the climate is mild relative to its latitude, but despite this, the hedgehog population seems to be low [47]. The Natural History Museum and Botanical Garden in Kristiansand is situated beside the campus of the University of Agder, and this area is one of the main areas for hedgehogs in the city [48]. The hedgehog became red-listed in Norway in 2021, and the aim of this study was to shed some light on the home range size, movements, and nest site selection of hedgehogs in a typical urban residential area in Kristiansand. By radio-tracking hedgehogs, we aimed to identify the challenges they face and the habitats that are especially important for their existence. We performed this study in late summer and autumn in order to identify where and when the hedgehogs chose to hibernate.

## 2. Methods

### 2.1. Study Area

This study took place in the Gimle/Lund area in the city of Kristiansand in Southern Norway (58.15° N, 8.00° E, Figure 1). Kristiansand has a population of 115,000 and covers an area of almost 644 km^2^. There are approximately 53,000 residences, and the city has a population density of 186 citizens/km^2^. Kristiansand is a coastal city that borders Skagerrak with relatively mild winters, given the latitude. The residential areas where this study took place are a mix of regular single-family homes with gardens, terraced houses, and apartment blocks. The terraced house gardens often connect to a larger communal lawn or park. Gardens are often divided by wooden fences, chain-link fences, and/or hedges, and most houses have open driveways without gates. The campus of the University of Agder Gimlemoen and the botanical garden, adjacent to the residential areas, have large, open lawns together with flower beds and bushes. Along roads and outside scattered office buildings, there are often beds of densely planted dense bush/hedge. In addition, hedgehogs, the most common mammals observed in the area are the Eurasian red squirrel (*Sciurus vulgaris*) and the brown rat (*Rattus norvegicus*). Badgers (*Meles meles*) are present but rarely reported or observed.

### 2.2. Citizen Science Initiative

The hedgehog population in the city of Kristiansand was mapped in 2019 using reports from the public [48]. This provided information that the Lund area, with the university campus and botanical garden, was one of the main areas where hedgehogs were observed. As preparations for the radio marking of hedgehogs from the 15 August 2022, we initiated a new citizen science campaign in June 2022 to obtain detailed information on where to find hedgehogs in this area, using radio stations, newspapers, Facebook groups, and posters along the roads. We received 45 observations of hedgehogs from the public. Every citizen hedgehog observation reported from this campaign was registered in the Norwegian Species Observation System [47].

### 2.3. Radio-Marked Hedgehogs

The field work lasted from 15 August to 5 November 2022 (Table 1). The registrations reported by citizens were used as a guideline when searching for hedgehogs with flashlights in the evenings. When a hedgehog was spotted and hand-captured, we determined the sex and weighted it by placing it inside a plastic box on a scale (max = 5000 g, d = 1 g). We had in total six radio-transmitters (R1680 glue-on transmitter, Advanced Telemetry Systems (ATS), Isanti, MN, USA) available, each weighing 3.6 g, including the 20 cm antenna, less than 0.4% of the body weight of any adult hedgehog. To fit the transmitter in a way that would not inhibit a hedgehog’s normal life, spikes in a concentrated area on the lower back were clipped 1–2 cm using scissors and clippers (as in [23,38,49]). The transmitter was carefully glued onto the clipped spikes and to the spikes next to them using epoxy glue, making sure that no glue touched the skin of the animal, and was firmly held in place for 10 min until the glue had cured properly. The transmitter was positioned with the antenna sticking out the back, so that it trailed behind the animal when it moved. This position enabled the animal to move freely without the transmitter effecting movement in any way. Of the originally six radio tagged hedgehogs, one was soon hit by a car, and two were killed by an unknown predator or died of other causes and scavenged after death. The three salvaged transmitters were therefore placed on three new hedgehogs as soon as they were located. Three of the six transmitters lost the signal after a while, probably due to transmitter failure. Thus, in total, nine hedgehogs were tagged and tracked, but at the end of the tracking period, only three hedgehogs were still equipped with a working transmitter (Table 2).

### 2.4. Radio Tracking

We used a scanning receiver (R410, ATS, USA) in conjunction with a smaller direction-based H-antenna (ATS, USA) and a much larger, five-element foldable Yagi Antenna (Model 17734, ATS, USA) to detect tagged hedgehogs. Tracking was carried out by car and by foot, depending on terrain, and the coordinates of every localization (or fix) were registered using a GPS. In the first period, from 15 to 30 August, the entirety of the night was spent looking for hedgehogs for tagging. A night typically started around 22:30–23:00 and lasted until 06:00–07:00. During the second period between 30 August and 13 September, a three-night rotation was performed, alternating between early half-night (22:00–23:00 to 03:00–03:30), late half-night (from 02:30–03:00 to 06:00–07:00), or full night (22.00–23.00 to 06.00–07.00, see Table 1). During all nighttime tracking, we rotated between all tagged individuals, and it usually took 1–2 h between every localization of an individual, depending on the number of hedgehogs marked at a given time.

After 13 September, a second phase of the project started to establish where and when the hedgehogs went hibernating. Between 24 September and 29 October (Table 1, periods 3 and 4), the individuals were tracked during the daytime. Since no nest site switches were observed after 29 September, we assumed hibernating was initiated, and to confirm this, we tracked once every night between 23 and 29 of October (Table 1, period 5). As there was no sign of hedgehog activity, neither day nor night, we ended the tracking on 5 November (Table 1, period 6). When hedgehog nest sites were investigated during the day, we noted the nesting type by category: Building (garage, porch, terrace, stairs, building materials) or Nature (vegetation, bush, forest). When the actual nest could be observed, we noted the nest materials as well.

Permissions to capture and tag hedgehogs were provided by the Norwegian Food Safety Authority (FOTS ID 27113) and the Norwegian Environment Agency (ref. 2022/7181), and permission to operate VHF tags was given by the Norwegian Communications Authority (PMR-no. 17808).

### 2.5. Data Analysis

All analyses were performed using R [50]. We estimated home-range size by both Minimum Convex Polygon (MCP) and Kernel Utilization Distribution (KUD) estimators, following Riber [51] using the sp [52] and adehabitatHR packages [53]. To examine to what extent home range size estimates were sensitive to localization outliers, we performed an incremental analysis by calculating MCP sizes based on 50 to 100 percent of localizations (with 5 percent increments) from each individual and plotted the relationship between the number of localizations and the estimated MCP area. We also calculated 95% and 50% of the kernel utilization distribution (KUD) area, where the smoothing parameter was set to 46.2, based on the reference bandwidth method. The 95% KUD excludes 5% of the most extreme localization outliers in order to better represent the “true home range”, while the 50% KUD is expected to represent the core of the home range [51]. As suggested by Seaman et al. [54], we only included individuals with more than 30 localizations when calculating home range sizes. We used a simple two-sided *t*-test to investigate if kernel home range sizes were different between sexes and linear regression to see if kernel home range sizes were affected by initial body size. We included (MCP) as this is used in many other relevant studies and therefore is good for comparison [22,24,33,35,36,38,39,51], but also KUD since there is a debate regarding which home range estimator is least biased [55,56].

All localizations of the six individuals between 22.00 and 07.00 were used to investigate the distance moved at night as a function of time of night and sex. The time in hours and the distance in meters between two successive localizations of the same individual on the same night were calculated. The distance was calculated as the shortest distance between the two localizations, ignoring physical structures like buildings, fences, etc., by using the distHaversine function in the geosphere R-package [57]. The distance in meters, divided by the time in hours, was used as the response variable. This variable was highly right-skewed, with many values being equal to or close to zero but with some extreme values, and therefore we applied a negative binomial mixed model, with individual ID as a random intercept (to account for multiple observations of the same individual), using the glmer.nb-function in the lme4-package [58]. We investigated the effect of the explanatory variables sex, as a binomial variable, and time in hours, relative to 22.00 (the earliest time of radio tracking at night), as a continuous variable. We started with the full model, including sex, time, and their interaction, and fitted all nested simpler models, including the null model. We used Akaike’s Information Criterion, corrected for small sample size (AICc) [59], to compare the fit of the different models. We also calculated the minimum distance moved per night per individual by adding all consecutive distances between pairs of localizations. This variable was highly right-skewed, with many values being close to zero, and therefore we applied a logistic regression with individual ID as a random intercept (to account for multiple observations of the same individual), investigating the effect of sex as a binomial variable on total meters moved per night. The model took into account that the number of pairs of localizations varied between nights and individuals.

## 3. Results

### 3.1. Radio Tracking

Nine hedgehogs (4 females ♀, 5 males ♂) were tracked during the field campaign, for a total of 482 localizations (see map in Figure 2 and Table 2). During this period, individuals 1, 2, and 4 died (see methods), and individuals 6 and 7 lost their radio transmitters after a few days. The transmitters were all recovered and then reused to mark other individuals. This allowed us to track nine different hedgehogs in total, despite only having six radio transmitters. However, due to being tracked only for a few days and thus not being localized many enough times, individuals 2, 6, and 7 were removed from the spatial analysis. At the time of tagging, the weights of the hedgehogs ranged from 928 g to 1410 g (Tabel 2), and the average weight of the females and males was 1007 g and 1271 g, respectively.

### 3.2. Home Range

Pairs of hedgehogs had overlapping 100% MCP home ranges with either the same or the opposite sex (Figure 2). In the middle of the area, two male home ranges overlapped (Male 5 and Male 9). Male 7 was also found in this area, as were males 5 and 9, but its transmitter fell off after only three days. In the south-eastern part of the area, two females had overlapping home ranges (Female 1 and Female 8). The size difference in home range between sexes was neither significant for 95% (t = 0.08, df = 4, *p* > 0.05) nor 50% (t = 0.5, df = 4, *p* > 0.05) kernel sizes (Figure 3, Table 3). There was no significant relationship between weight at capture and home range size at neither 95% (F (1,4) = 0.05, *p* > 0.05) nor 50% (F (1,4) = 0.9, *p* > 0.05).

### 3.3. Movement

The distance moved was analyzed based on 196 pairs of localizations from the six individuals. The number of pairs per individual per night varied between 1 and 7 (median = 3). The time between two localizations of the same individual varied between 0.29 and 3.54 h (median = 0.87) and the distance moved per hour varied between 0 and 420.5 m/h (median = 42.0). Even with such large overall variation in distance moved, this variation was high within most individuals (Figure 4A), and only 21% and 7% of the variation were explained by individual differences in the full model and null model, respectively. There were some differences in median distance moved between the six individuals, with individual numbers 3 (a male) and 4 (a female) standing out as individuals showing a low and high median distance moved, respectively (Figure 4A). However, there was no clear tendency between the other four individuals for a systematic sex effect, and the AICc value of the sex model (Table 4, model 4) was lower than that of the null model (model 3), indicating low support for a systematic difference between sexes in distance moved per hour. There were clearly more longer distances moved per hour during the last part of the night, especially between 04:00 and 06:00, and these longer bursts were more common in some individuals than others, but there were also many observations with no movement between two successive localizations during the whole night (Figure 4B). The model with only the time effect was the one with the lowest AICc value (Table 4, model 1), but this model was not substantially better than the null model (deltaAICc = 1). All in all, the model selection shows that there is low support for any sex difference and some support for an increase in movement per hour at the end of the night.

The total distance moved during a single night for one individual varied between 0 and 667 m. Despite both females and males being localized on average an equal number of times per night (mean pairs of localizations per night = 2.48 and 2.53 for females and males, respectively), and the fact that the maximum distance moved was more or less similar for both sexes (Female 4: 605 m and male 9: 667 m), female total movement per night was overall higher than that of males (Χ^2^ = 4.65. df = 1, *p* = 0.03). The median number of meters moved per night for females was 227, and the median for males was only 70, and this difference was largely due to the fact that males often did not move, or moved very little, between pairs of localizations. However, the variation within sexes was large, and *p*-values from mixed logistic regression models must be interpreted with some caution [60].

### 3.4. Nest Use

The six hedgehogs used a total of 28 different nest sites (see Table 5); 18 of these nests were used by males and 10 by females. The number of nest sites used per individual varied from one to 10, and the maximum of 10 nest sites was that of a male, and the second of seven nest sites was that of a female. The individual hedgehogs switched nests from zero to 14 times during the tracking periods; the maximum of 14 switches was both by a male and a female. Three females did a total of 15 switches, and four males did 21 switches, so the total number of nest switches was 36. No hedgehog used the nest of another hedgehog. Approximately 16 of the 28 nests were under/inside a building (garage, porch, veranda), and 12 were in natural habitat (hedge, bush, forest). Nest sites of six hedgehogs are shown in Appendix A.

Only three individuals remained tagged with a radio transmitter until hibernation: female nr 8; male nr 3; and male nr 9, and these individuals did not have overlapping home ranges. Male nr 3 stayed in the area around the campus and botanical garden at the University of Agder. The first observations were made in a residential area south of the university campus, but later they moved to campus and went hibernating under the root of a pine tree (*Pinus sylvestris*) in a small patch of natural wood with pine trees, scrub, heather, and tall grass. The 15th of September was the last time the hedgehog switched nests, and we suppose that he went hibernating. Female nr 8 moved back and forth between nest sites under buildings by crossing a road that has high traffic during the daytime. The 29th of September was the last time this hedgehog switched nests, and it established itself in a small patch of natural habitat with blackberries (*Rubus* sp.), under the root of a big deciduous tree, next to a rock, and we assume that she went hibernating from this date. Male nr 9 was moving between a nest in a private garage and a nest in a field of planted bushes along a road. There was no nesting material under these bushes, apart from a few small, dry leaves and a little trash. For hibernation, he found a patch of planted yew (*Taxus baccata)* along a road where he made a burrow under the roots. The 16th of September was the last time this hedgehog switched nests. All three hedgehogs chose hibernation sites under tree roots in as many natural environments as they could possibly find.

### 3.5. Nesting Material

We observed the nesting material in 14 of the 28 nests, belonging to six of the individuals (Table 5). The nesting material was primarily sourced from locally available vegetation, but two nests consisted completely or partially of plastic and paper trash, and at one nest site, the hedgehog slept openly in a flowerbed. One nest consisted only of grass, two consisted of pine/yew needles mixed with trash, two consisted of leaves, and eight consisted of a mix of grass, leaves, and/or moss. Two of the three hibernating hedgehogs found sufficient nesting material in the surroundings in the chosen small patches of woodland, but the last hedgehog (in a planted yew-bed) only found yew needles and some trash as nesting material. We were worried that this would not be sufficient protection during the winter, but fortunately this individual managed to dig down under the roots after a few days, and we hope that this was enough for it to survive the winter.

## 4. Discussion

### 4.1. Home Range

To investigate hibernation nest sites, our data were collected in late summer and autumn. We did not find a statistical difference in the home range size between the two sexes in our study at this time of year. This lack of difference could be due to the low sample size in our study, as the variation in estimates was great, especially among males. In Italy, Bottani and Reggiani [33] did not find any significant sex difference in home range size, but that home range sizes varied greatly (more than 10-fold), with the largest being more than 100 ha. Although we did not detect home range sizes near this size, we also found substantial variation in home range sizes for both sexes (Figure 3, Table 3). Reeve and Morris [61] found that male hedgehogs on a golf course in England had larger home ranges and faster and longer movements than females, and both Reeve and Morris [61] and Bottani and Reggiani [33] report a mating season from May until September, which is different from the seasonality in home range size between the genders that we find in the Nordic countries.

Male and female home ranges of similar sizes late in the season, as we found in our study, are also described elsewhere in Fennoscandia. Close to the northern boundary of the species’ distribution, hedgehogs seem to terminate the mating season earlier, and there is a seasonal change in home range sizes and movements, with males having the larger home ranges and traveling the longest in spring/early summer and with less or no sex difference in autumn [26,36,38]. The study area of our choice in Kristiansand, Norway (58° N) is comparable to that of Rautio, Valtonen, Auttila, and Kunnasranta [38] and Rautio, Valtonen, Auttila, and Kunnasranta [41] in Joensuu, Finland (63° N), though Joensuu is situated even further north and east in Fennoscandia and thus probably has a somewhat colder climate. Rautio, Valtonen, and Kunnasranta [38] found that, in the pre-hibernation period, adult males and females had a home range of 17 and 29 ha, respectively, and our home range estimates are of similar size (95% KUD ranges from 11 to 22.2 ha). Also, the body weight of the hedgehogs in the pre-hibernation period was very similar in Finland and Norway, with males being heavier than females. In Kristiansand, the male average weight was 1272 g (*n* = 5) and female 1007 g (*n* = 4), and in Finland, the male average weight was 1286 g (*n* = 5), females 958 g (*n* = 4) [38]. Food availability within the home range might affect home range size, and the availability of food in urban and wild environments might be quite different [28]. Uneven food availability within the study area might be the reason for the individual differences in home range size that we have observed, but unfortunately, we do not have data to test this.

The precision of home range size estimates is sensitive to the sample size [54]. In Finland, Rautio, Valtonen, and Kunnasranta [38] showed that less than 30 localizations from individual hedgehogs should be enough to provide unbiased home range sizes, and Pettett, Moorhouse, Johnson, and Macdonald [22] found that 20 localizations were sufficient. However, our incremental analysis indicates that we, with as many as 50+ localizations, do not have a sufficient sample size to ensure unbiased estimates (Appendix A). Indeed, for some of our individuals, the localizations are quite uniformly distributed within the home range, with no clear core areas (see individuals 4 and 5 in Figure 2), while others have two core areas and/or MCP home ranges that are clearly affected by one or a few outliers (see individuals 2 and 9 in Figure 2). We therefore expect our estimates to be somewhat negatively biased.

We found that our hedgehogs in Kristiansand of both the same and opposite sex had overlapping home ranges, which support the findings in a large number of studies (see, e.g., Campbell [34]; Parkes [37]; Reeve [39]; Bottani & Reggiani [33]; Kristiansson [36]; Riber [51]; Rautio et al. [38]). Studies in our neighboring countries, Finland and Denmark, have shown that females tend to have fewer overlapping home ranges and core areas than males during late summer and autumn, as mutual avoidance can ensure enough food availability to increase fat deposits before winter [38,51]. In Ireland, Haigh et al. [62] found that each hedgehog occupied a distinct area of the arable field and rarely crossed the path of another. Cassini and Föger [63] found that hedgehogs showed mutual avoidance and suggested that this imposes a limit on the number of animals in an area. We did not observe any hedgehogs at the same spot as another hedgehog in our study, which might indicate that they are solitary animals practicing mutual avoidance in time, despite having, at least partially, overlapping home ranges.

Two of the marked hedgehogs were found dead and scavenged 3 and 16 days after marking, respectively, and although we could not establish the cause of death, it is possible that these two individuals were killed by a predator. Predators are generally very rare in the study area. The presence of badgers, however, has been shown to restrict hedgehog movement and foraging and lead to smaller home ranges [22,64]. Badgers have been observed within our study area, but rare reports and the fact that we did not observe any during this study suggest that direct or indirect interactions between badgers and hedgehogs were uncommon. None the less, it is possible that some of the variation in space use among the hedgehogs included in our study can be explained by the local presence of one or more badgers.

### 4.2. Movement

We did not find any significant difference in the distance moved per hour between the sexes. Both males and females moved as much as 300–400 m/h, but this was not common, and we also observed individuals (mostly males) not moving at all between successive localizations at any given time of night. We believe that the timing of our study, the pre-hibernation period, August to September, is the explanation for this observed pattern. Adult females may use longer time to feed and build up a fat layer before winter after reproduction in summer and therefore need to move more actively around in the pre-hibernation period in order to locate enough food, while males have had longer time to feed since the mating period in the spring. This is partially supported by Kristiansson [36], who found that males in Sweden traveled longer than females in the mating season but that males and females traveled similar distances in the post-mating season.

Based on this, we would expect similar distances between males and females, but in our study, both sexes moved considerably shorter than what has been reported from Sweden [36]. This difference in distance moved between two otherwise quite similar climatic regions may depend on both the nature of the habitat, food availability, and the time interval and length of time of the registration points. The frequency of registrations during the night was higher in the study of Kristiansson [36] with locations every 15 min, while we usually had more than one hour between our registrations and sometimes several hours. We expect that these longer intervals will lead to a higher degree of underestimation since this method only registers the shortest distance between two localizations. In addition, Kristiansson [36], did his study in a small town that could be considered rural. Further, it has been suggested that there is higher food availability in urban environments compared to rural environments [9], and this could lead to hedgehogs in our study not having to move as much to find sufficient food. Hedgehogs spend most of the night in the post-mating season to forage [29,35,36], and we therefore suppose that the shorter distance traveled in our study is related to easier access to food. On the other hand, a study in an urban environment in the UK [24] found much higher travel distances than in our study, but gardens in our study were mostly open and easy to access, while urban gardens in the UK are usually smaller and harder to access [65], and this could result in the hedgehogs having to travel longer distances to find sufficient food.

We detected no active hedgehogs before 22:00, and most individuals left the nest around 23:00, when the neighborhood traffic, by car or foot, was strongly reduced. Almost all anthropogenic disturbances in this area had ceased by midnight. Dowding, Harris, Poulton, and Baker [24] found that hedgehogs were more active after midnight as a risk-reducing behavior. Based on this, we suspect that the hedgehogs in our study may have started their nightly movements as late as 23:00 to minimize exposure to human activity. Most of our hedgehogs increased their average movements after 03:00, and this was the portion of the nights with minimum disturbances. We did observe that the hedgehogs consistently began moving towards their nests from around 03:00, and thus it seems that this increased movement in the late part of the night is linked to the need to find shelter before dawn when human activity increased.

### 4.3. Nests

We did not find any evidence of nest sharing in our study, and this might be the normal pattern of nest use by hedgehogs. Nest-sharing simultaneously in wild-living hedgehogs is rare, though non-simultaneous nest-sharing has been documented [32,41,51,61,62,66]. Nest sharing can increase transmission of ectoparasites [41,61], and this could be the reason why nest sharing is rarely observed among hedgehogs. Ectoparasite exposure might also explain why hedgehogs alternate between several nests, as this can reduce parasite exposure (see, e.g., Stanback and Dervan [67]; Bize et al. [68]). However, we never observed ectoparasites in our study, suggesting that ectoparasite burdens overall were low.

In total, we found 28 different nest sites used by seven hedgehogs (the remaining two individuals were never observed in a nest during the short period they were tracked), and 18 and 10 nests were used by males and females, respectively. Three males and three females switched nests a total of 21 and 14 times, respectively. Bottani and Reggiani [33] in Italy and Reeve and Morris [61] in England found that females used the same nest repeatedly for periods significantly longer than the males, and that this was due to reproduction or because the males, with larger home ranges, used several nests rather than having to move long distances to reach the same one. In our study, there were no clear differences between sexes, neither in the number of switches nor the number of nests used, but large differences between individuals overall (Table 5), and this is in accordance with several other studies [35,41,61,69].

The nest is a very significant feature in a hedgehog’s life, particularly during hibernation [42], determining both its habitat choice and distribution. The nest location habitat has been studied in several countries: the UK ([22,24,42,61,62], New Zealand [69], Denmark [26,44,51], and Finland [41]. The most common nesting habitat is hedgerow or forest [22,51,62,70,71] and forest is especially preferred for hibernation [44,70,72]. In Finland, Rautio et al. [72] observed that the hedgehogs in urban environments still preferred to hibernate under tree roots in forest patches, exactly as we observed for our three hedgehogs in Kristiansand. Such habitats are usually limited in urban environments, and our result indicates that maintenance of natural forest patches in urban areas where hedgehogs exist may improve habitat suitability, and thus survival, and help mitigate the present decline in hedgehog numbers. Hedges and hedgerows, often common in urban environments, are known to be of special importance for hedgehogs during the active season since they offer shelter, nest locations, and food [22,25,51,62,70,71,73]. We also observed that hedgehogs were often found in hedges, both during their nightly movements and when sleeping during the day. Manmade constructions, such as playhouses, sheds, porches, and terraces, were also often used as nest locations.

Leaves or grass are known to be the two most important nesting materials for hedgehogs [22,26,41,42,44,51,61,71,73], and we also found that leaves and grass were most often used, but that hedgehogs used other material available close to the nest site, regardless of source. Nest sites under dense hedgerows, which do provide protection from predators, did not necessarily offer sufficient nesting material for hibernation, and the hedgehogs abandoned such habitats and rather utilized the more limited forest patches when preparing for hibernation. We worry that lack of proper nesting material and limited availability of forest patches may increase winter mortality in urban environments, especially in more northern regions where winter temperatures can be more challenging.

The hedgehogs in our study went hibernating in September, comparable to what is described in Finland [41], but much earlier than what is common in other places in Europe [33,62]. We do not know what initiated the start of hibernation in our study, but we assume that a combination of sufficient energy storage and external zeitgebers triggered this. The fact that hedgehogs in Finland, with comparable body weights, also initiated hibernating in September suggests that this is the normal behavior of hedgehogs in the extreme north of the species distribution in Europe.

## 5. Conclusions

This study offers new knowledge of hedgehog home range sizes, movements, and nesting behavior in suburban areas close to this species’ northern boundary. Home range size, distance moved, and the number of nests varied greatly between individuals, and no sex effect was observed. Our results indicate that hedgehogs in urban environments prefer forest patches and use natural nesting material when preparing for hibernation, and this supports the findings from other studies in Fennoscandia. We therefore conclude that it is important to maintain existing and establish new forest patches in urban environments to ensure hedgehog winter survival in the north. It would also be beneficial to encourage garden owners to contribute by providing shelters with ample natural nesting material in their gardens.

## Figures and Tables

**Figure 1 animals-14-00130-f001:**
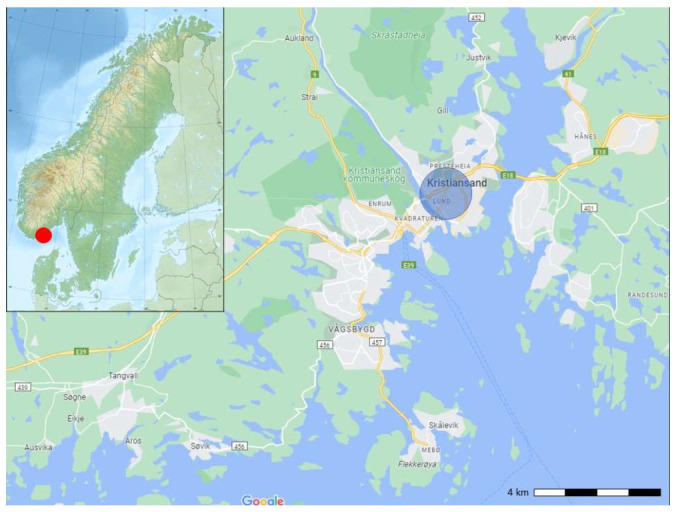
Map showing the location of Kristiansand in Norway (red dot), the rural areas (green) the densely populated areas (light grey) and a blue circle surrounding the Lund area with the University of Agder and The Natural History Museum and Botanical Garden where the study was conducted. (Map sources: © Google, 2023, and © NordNordWest/Wikimedia Commons/CC-BY-SA-3.0).

**Figure 2 animals-14-00130-f002:**
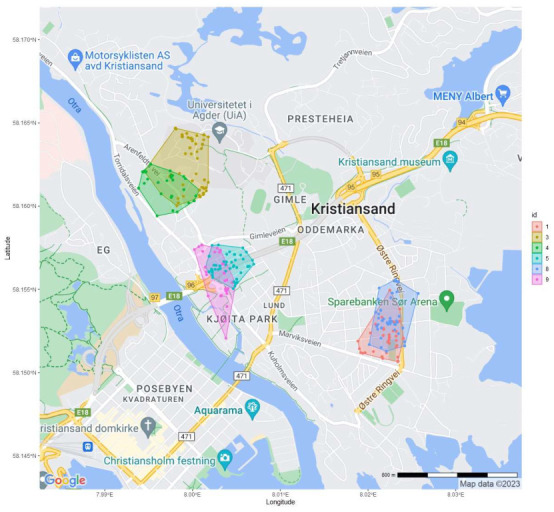
Map of Lund and Gimle in Kristiansand (Southern Norway) with 100% minimum convex polygon (MCP) home ranges of six hedgehogs, displayed with color-codes for each individual (1, 4, 8 = females, and 3, 5, 9 = males). The home ranges overlap both within and between sexes.

**Figure 3 animals-14-00130-f003:**
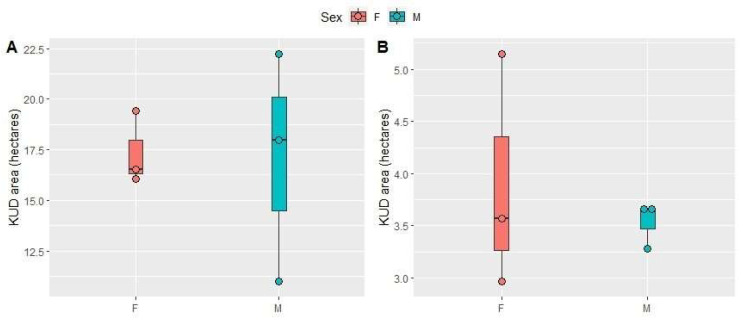
Approximately 95% (**A**) and 50% (**B**) Kernel Utilization Distribution (KUD) estimates, in hectares, of three male (blue) and three female (red) adult hedgehogs in the pre-hibernation period. Points represent individual estimates, and the horizontal line in the box represents the median.

**Figure 4 animals-14-00130-f004:**
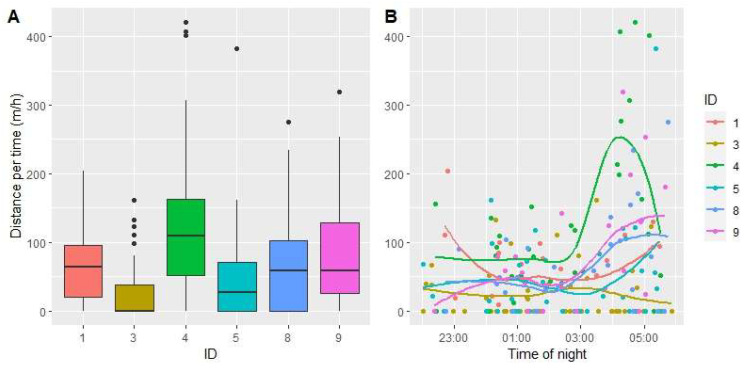
Distance moved (m/h) between successive relocations the same night, as a box plot with each individual (**A**) and as a function of time at night for each individual (**B**). Time in (**B**) represents the first of the two successive localizations. Lines are created by loess-smoothing.

**Table 1 animals-14-00130-t001:** Overview of the six field periods, with start and end dates in 2022, type of field activity (M = marking individuals, T = radio tracking individuals), and the number of tracking sessions in each period. In period 2, the time alternated between early and late night. The shading of the cells with time indicates light conditions. Dark gray = nighttime, light gray = dusk, and white = daytime.

Period	Start Date	End Date	Time of Day	Field Activity	No. Days
1	15 August	30 August	22.30–07.00	M + T	16
2	31 August	13 September	22.00–03.30/02.30–07.00	M + T	14
3	24 September	10 October	10.00–14.30	T	17
4	11 October	20 October	17.00–20.00	T	9
5	23 October	29 October	00.00–02.00	T	4
6	5 November	5 November	11.00–14.00	T	1

**Table 2 animals-14-00130-t002:** Detailed information on each individual hedgehog tracked, including sex, body mass, start and end of tracking, number of days tracked, and number of localizations (individual registrations) during the period between 15 August and 5 November 2022.

ID nr.	Sex	Body Mass (g)	Date Tagged (Start Tracking)	Last Tracking	Cause of End	Days Tracked	Localization
1	Female	958	15 August	30 August	Road killed	14	49
2	Female	1044	22 August	19 August	Predation	2	4
3	Male	1361	16 August	5 November	End of campaign	27	124
4	Female	928	17 August	3 September	Predation	17	62
5	Male	1244	17 August	9 September	Lost signal	22	88
6	Male	970	18 August	20 August	Lost signal	2	4
7	Male	1410	22 August	24 August	Lost signal	3	9
8	Female	1099	27 August	5 November	End of campaign	16	84
9	Male	1374	4 September	5 November	End of campaign	9	58

**Table 3 animals-14-00130-t003:** Approximately 50% and 95% kernel utilization density (KUD) home range estimates, as well as 100% maximum convex polygon (MCP) home range estimate (in hectares), for each of the six individuals, based on all localizations.

ID	95 KUD	50 KUD	100% MCP	Sex
1	16.0	3.6	8.8	F
3	22.2	3.7	15.6	M
4	19.4	5.1	9.0	F
5	11.0	3.3	6.3	M
8	16.5	3.0	10.7	F
9	18.0	3.7	8.7	M

**Table 4 animals-14-00130-t004:** A list of negative binomial mixed effects models was used to investigate the effect of sex and time of night and the two-way interaction (Time:Sex) on the distance moved by hedgehogs between successive localizations. The inclusion of a term or model is indicated by an “x”. AICc represents Akaike’s Information Criterion (a lower value indicates better model fit), and deltaAICc represents the difference in AICc-value between the current model and the model with the lowest AICc-value (model 1).

Model	Time	Sex	Time:Sex	Df	AICc	deltaAICc
1	x			4	1785.9	0
2	x	x		5	1786.4	0.5
3				3	1786.9	1
4		x		4	1787.5	1.6
5	x	x	x	6	1787.8	1.9

**Table 5 animals-14-00130-t005:** Nest use by the seven hedgehogs of which this was documented. The table shows the total number of nests (Nests) and the number of nest switches (Nest switches), as well as the nest location and nest material when this was possible to observe. Numbers in parenthesis in nest location and nest material represent the number of nests observed per individual. A date of hibernation is provided for those three individuals that were monitored until this happened. Explanation of abbreviations: Nest location: B = Building, V = Vegetation. Nest material: P = Pine needles, T = Trash, L = Leaves, G = Grass, M = Moss, Y = Yew needles, and N = No material.

Individual	Nests	Nest Switches	Nest Location	Nest Material	Date Hibernation
F1	2	1	B (2)	-	-
M3	10	14	B (5), V (5)	P (1), T (1), L (1), G/L/M (2)	12 September
F4	1	0	V (1)	L (1)	-
M5	3	2	B (3)	G/L (2)	-
M7	1	0	B (1)	G/L (1)	-
F8	7	14	B (4), V (3)	G (1), G/L (2)	29 September
M9	4	5	B (1), V (3)	N (1), Y/T (1)	16 September
TOTAL	28	36			

## Data Availability

The data presented in this study are available on request from the corresponding author.

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
