# Peer review of "Home Range, Movement, and Nest Use of Hedgehogs (Erinaceus europaeus) in an Urban Environment Prior to Hibernation"

_animals, 2023, doi:10.3390/ani14010130_

Round 1

Reviewer 1 Report

Comments and Suggestions for Authors

 I would like to make some suggestions to improve the quality of the paper as below:

Line 11: change ‘simple summary’ to ‘Summary: 

Lines 11-23: Please rephrase this sentence.

Summary

At the end of the summary section, authors may also say in 1-2 sentences that their findings contribute to the conservation of species and its habitat. And authors can show the significance of the study to local forest ecosystem protection in 1 sentences. 

Lines 24-37: Please rephrase this sentence.

Abstract

I think the abstract needs to be rephrased and improved. In my opinion, it is good to start with the problem examined in the study. Within this context, the main problem that is examined by the authors should be explained in 1-2 sentences at the beginning of the abstract. After that, the methods(specific statistical model)and the main and detailed results should be given briefly. This can be followed by the main findings of the study. Finally, what is the importance of the results and how the results contribute to further studies should be written down. In my opinion, it is always good to finish the abstract with such a sentence. 

Lines 137-149:Please rephrase this sentence.

Methods:

I think, as an article discussing mammal (Erinaceus europaeus) population conservation,The authors describe and state that the remaining species in the study area are necessary.In particular, species and factors that may have an impact on the behavior and population of study subjects. 

Discussion:

The authors mention that Mammals that are common in urban environments  include the badger (Meles meles), red fox (Vulpes vulpes), brown rat (Rattus norvegicus) and so on.So whether the impact of these species on the range and nesting situation of hedgehogs needs to be discussed and analyzed.In addition, the authors show that after November, hedgehogs enter a hibernation period.Before that, did food resources have an impact on their range of activities and nesting locations?There are differences in the availability and weight of food in urban and wild environments. Whether to discuss. 

Conclusions:

The limitations of the study should be given in the conclusion section.

Author Response

Response to reviewer 1

Thank you for your valuable input to the manuscript. Below you will find our detailed answers to your comments.

Line 11: change ‘simple summary’ to ‘Summary:’ 

As far as we understand, the journal template defines the structure, including the title “Simple Summary”. We will leave this as it is, but if we have misunderstood, we expect the editor will make sure this is changed.

Lines 11-23: Please rephrase this sentence.

Thank you for your advice, we have rephrased, and shortened the simple summary.

Summary

At the end of the summary section, authors may also say in 1-2 sentences that their findings contribute to the conservation of species and its habitat. And authors can show the significance of the study to local forest ecosystem protection in 1 sentences. 

Lines 24-37: Please rephrase this sentence.

Thank you for valuable comments, we have rephrased and tried to improve the Summary according to this.

Abstract

I think the abstract needs to be rephrased and improved. In my opinion, it is good to start with the problem examined in the study. Within this context, the main problem that is examined by the authors should be explained in 1-2 sentences at the beginning of the abstract. After that, the methods(specific statistical model)and the main and detailed results should be given briefly. This can be followed by the main findings of the study. Finally, what is the importance of the results and how the results contribute to further studies should be written down. In my opinion, it is always good to finish the abstract with such a sentence. 

Lines 137-149:Please rephrase this sentence.

We agree that the abstract needed attention. We have rewritten it, based on your suggestions.

Methods:

I think, as an article discussing mammal (Erinaceus europaeus) population conservation,The authors describe and state that the remaining species in the study area are necessary.In particular, species and factors that may have an impact on the behavior and population of study subjects. 

We have included some information of the other commonly observed mammals in the area red squirrel and brown rat, in the end of section 2.1 (l. 135-138)

Discussion:

The authors mention that Mammals that are common in urban environments  include the badger (Meles meles), red fox (Vulpes vulpes), brown rat (Rattus norvegicus) and so on.So whether the impact of these species on the range and nesting situation of hedgehogs needs to be discussed and analyzed.

As badgers, the only of these species present and also expected to have an impact on hedgehogs, were very rare in the area, we had chosen not to include this in the discussion, but we appreciate the advice and have included a section of this in the discussion (l. 448-454).

In addition, the authors show that after November, hedgehogs enter a hibernation period.Before that, did food resources have an impact on their range of activities and nesting locations?There are differences in the availability and weight of food in urban and wild environments. Whether to discuss. 

Our hedgehogs went hibernating in September, but since we do not have data on food availability we cannot be certain of what initiated the hibernation. Since hedgehogs in Finland also went hibernating early as in Norway we suspect that this is the normal behavior of hedgehogs up north. We have tried to address this in the discussion (l.543-549).

Conclusions:

The limitations of the study should be given in the conclusion section. Thank you for the suggestion. We have rewritten the conclusion according to your suggestions.

Reviewer 2 Report

Comments and Suggestions for Authors

General recommendations
The manus concerns the home range, movements and nesting behaviour of urban hedgehogs. The results and conclusions are based on few data. However, these data are ads important knowledge to the biology of hedgehogs that are not easy to receive. In general, the paper should be shorter and written in a less report like style e.g. do not use unnecessary words. And please remove speculations and genereal management suggestions that is not suported by your data. Do not present tables with almost the same content (Table 1 and 2). I suggest that you delete table 1. It does not ad necessary knowledge to discuss. Please check the biology of hedgehogs-  to the best of my knowledge hedghogs are very active during late summer and autumn when this study is conducted – females may have young at that time of year as young are born during August or September. Hedgehogs mate in mid-summer. And the time of day that hedgehogs are active are related to sunset and during autumn when hedgehogs prepare for hibernating and have to gain fat for the long winter they are active long before midnight. Helge Walhovd has written some very comprehensive papers on both the breeding biology and the hibernation and nesting behaviour of hedgehogs. Also, Sofie Rasmussen has looked at the survival of young hedgehogs:

The breeding habits of the european hedgehog (Erinaceus europaeus L.) in Denmark. WALHOVD, H. Zeitschrift für Säugetierkunde, 1984, Vol.49 (5), p.269-277.

Partial Arousals from Hibernation in Hedgehogs in Outdoor Hibernacula. Walhovd, Helge. Oecologia, 1979, Vol.40 (2), p.141-153

The ecology of suburban juvenile European hedgehogs (Erinaceus europaeus) in Denmark

Rasmussen, Sophie L. ; Berg, Thomas B. ; Dabelsteen, Torben ; Jones, Owen R.

Ecology and evolution, 2019, Vol.9 (23), p.13174-13187

And I need a paragraph about Ethical statement about animal handling. Did you have a persission to handle the hedgehogs og is this not needed.

Suggestions to the text
Line 11: Insert population after hedgehog
Line 16: “The behaviour of males and females was equal…” This could be lack of data. Write:  No difference could be found between males and females….
Line 24: Insert population after hedgehog
Line 67: Please check your references carefully. I do not find any empirical evidence of foxes having impact on hedgehog populations. Foxes normally do not feed on hedgehogs and have coexisted for centuries in both Danish and English cities. Do not event new postulates. Badgers may occasionally eat hedgehogs but it is not a normal diet item for badgers.
69: Mention lawns in gardens as primary diet sources for hedgehogs here they find insects and slugs and snails. An old reference is: DW Yalden - Acta theriologica, 1976 “A better idea of what constitutes the diet of hedgehogs is given by estimating the weight of different food ingested”.
123: Please use the references of Helge Walhovd about hibernating Danish hedgehogs
177: What do you mean by “the antenna was trailed behind the animal? How long was it?
189: What do you mean by relocation- did you move the hedgehog- do you mean GPS localization or fix?
192: Please remove all the spaces – there a many
198-210: Please shorten this paragraph- much of this is not relevant for understanding the methods and for the discussion. Please combine table 1 and 2 they have many overlapping informations
280-283: The lack of significance may be doe to lack of data- this should be mentioned.
295: relocation means that you move the animal- find an other word- localization?
273: Do you know anything about how much nesting material hedgehogs use if they have access to nest material- do they collect it- I don’t think so.
385: Is it possible to make a table about these results- the 14 nests and which nest material was used.
402: “In Nordic counties it seems”- this is much to lose- find reference to this. To my knowledge hedgehogs mate in mid-summer and most give birth to young in late July an August.
409: Seems again- this is to lose. Just describe the two study areas compared.
507: I suggest you combine section 4.3 and 4.4 combine these two sections- shorten and focus. Why do hedgehogs need more nests? Do they have parasites like fleas?
585: Mortality- please delete this section -you have not studied mortality of hedgehogs- this is just speculation-
586-588. Conclusion- Do only conclude on what you have data to conclude. Focus on home range and nest material. Please no speculations of general management

Author Response

Response to reviewer 2

Thank you for your valuable input to the manuscript. Below you will find our detailed answers to your comments.

General recommendations
The manus concerns the home range, movements and nesting behaviour of urban hedgehogs. The results and conclusions are based on few data. However, these data are ads important knowledge to the biology of hedgehogs that are not easy to receive. In general, the paper should be shorter and written in a less report like style e.g. do not use unnecessary words. And please remove speculations and genereal management suggestions that is not suported by your data. Do not present tables with almost the same content (Table 1 and 2). I suggest that you delete table 1. It does not ad necessary knowledge to discuss. Please check the biology of hedgehogs-  to the best of my knowledge hedghogs are very active during late summer and autumn when this study is conducted – females may have young at that time of year as young are born during August or September. Hedgehogs mate in mid-summer. And the time of day that hedgehogs are active are related to sunset and during autumn when hedgehogs prepare for hibernating and have to gain fat for the long winter they are active long before midnight. Helge Walhovd has written some very comprehensive papers on both the breeding biology and the hibernation and nesting behaviour of hedgehogs. Also, Sofie Rasmussen has looked at the survival of young hedgehogs:

The breeding habits of the european hedgehog (Erinaceus europaeus L.) in Denmark. WALHOVD, H. Zeitschrift für Säugetierkunde, 1984, Vol.49 (5), p.269-277.

Partial Arousals from Hibernation in Hedgehogs in Outdoor Hibernacula. Walhovd, Helge. Oecologia, 1979, Vol.40 (2), p.141-153

The ecology of suburban juvenile European hedgehogs (Erinaceus europaeus) in Denmark

Rasmussen, Sophie L. ; Berg, Thomas B. ; Dabelsteen, Torben ; Jones, Owen R.

Ecology and evolution, 2019, Vol.9 (23), p.13174-13187

Thank you very much for your valuable comments and suggested references. We are glad to hear that you see the importance of this study, despite the low sample size. We have gone through the whole manuscript based on your general comments.

Regarding table 1 and 2: See comment further down.

Regarding suggested references: As we have not studied winter hibernation or reproduction directly, we chose not to focus too much on this, and the number of references related to this is therefore limited. However, Rasmussen et al (2019) was already used and we have, based on the suggestion, included Walhovd (1979) as well.

Regarding mating and nestlings: It seems, hedgehogs in Norway and Finland give birth earlier than in Denmark and UK etc, and we did not notice any nestlings or small hoglets in our study form August and onwards. However, this is an important aspect that should be addressed in future studies.

We have now removed speculations and management suggestions when it is not relevant.

We have also included a section on hibernation and body condition in the end of the discussion (l.543-549)..

And I need a paragraph about Ethical statement about animal handling. Did you have a persission to handle the hedgehogs og is this not needed.

This is stated in the manuscript, in the Methods chapter 2.4 Radio tracking, (l. 199-202).

Suggestions to the text:

Line 11: Insert population after hedgehog
Line 16: “The behaviour of males and females was equal…” This could be lack of data. Write:  No difference could be found between males and females….

Line 24: Insert population after hedgehog

Thank you for the suggestions. Based on suggestions from both reviewers we have rewritten the Simple Summary and the Abstract.

Line 67: Please check your references carefully. I do not find any empirical evidence of foxes having impact on hedgehog populations. Foxes normally do not feed on hedgehogs and have coexisted for centuries in both Danish and English cities. Do not event new postulates. Badgers may occasionally eat hedgehogs but it is not a normal diet item for badgers.

Thank you for point this out, as we fully agree that all statmenst in the manuscript should be based on sound knowledge.  We made changes to this, based on your suggestion. However, as badgers have been shown to have impacts on hedgehog behavior, we have included a section on this (l. 448-454).

69: Mention lawns in gardens as primary diet sources for hedgehogs here they find insects and slugs and snails. An old reference is: DW Yalden - Acta theriologica, 1976 “A better idea of what constitutes the diet of hedgehogs is given by estimating the weight of different food ingested”.

This is a relevant perspective. We have included this in the introduction and cited Yalden (1976) (l. 76-77).

123: Please use the references of Helge Walhovd about hibernating Danish hedgehogs
Thank you for directing us towards this publication. We have included this in the introduction (l. 108-111).

177: What do you mean by “the antenna was trailed behind the animal? How long was it?
We have rephrased the text in the methods in order to clarify this (l. 161 and l. 166-169).

189: What do you mean by relocation- did you move the hedgehog- do you mean GPS localization or fix?
The term “relocation” is used in some publications and is identical to the term “fix”. However, we acknowledge that this term can lead to confusion, and we have replaced this with the term “localization” throughout the manuscript.

192: Please remove all the spaces – there a many

This is done.

198-210: Please shorten this paragraph- much of this is not relevant for understanding the methods and for the discussion. Please combine table 1 and 2 they have many overlapping informations.
We have shortened the text in the paragraph as instructed (l. 189-198).
While table 1 presents when we were out tracking the animals, table 2 presents the data recorded for each individual hedgehog tagged. These two tables therefore present different information, and we find it hard to combine these. We think both should be included in the manuscript. However, we are willing to consider moving one or both tables (maybe table 2 is less critical to keep in the main part of the manuscript) to the supplementary material if this suggestion is supported by the editor.

280-283: The lack of significance may be doe to lack of data- this should be mentioned

Yes agree,and we have included this in the discussion (l. 389-392)

.
295: relocation means that you move the animal- find an other word- localization?

As suggested, we have replaced relocation with localization in the manuscript.

273: Do you know anything about how much nesting material hedgehogs use if they have access to nest material- do they collect it- I don’t think so.

We cannot find any reference to nesting material in line 273, so we suspect this line number reference might be wrong. None the less, we have no knowledge of whether hedgehogs collect nesting material or not, but we have cited Morris (1973, 2014) describing thickness of nesting material in winter nests (l. 104-106).

385: Is it possible to make a table about these results- the 14 nests and which nest material was used.

Thank you for this suggestion. We have included information on nesting material in table 5, which already included other information on nests, and we have made slight changes to the level of detail in section 3.5 accordingly.

402: “In Nordic counties it seems”- this is much to lose- find reference to this. To my knowledge hedgehogs mate in mid-summer and most give birth to young in late July an August.

Thanks for valuable comment, we have corrected this. In Norway, hedgehogs mate in May and give birth around midsummer, normally (data from 2 years of radio tracking hedgehogs in Trondheim, Johansen unpublished results)

409: Seems again- this is to lose. Just describe the two study areas compared.

Thanks again for your valuable comment. We have reorganized parts of section 4.1 and have then also changed our phrasing regarding this.

507: I suggest you combine section 4.3 and 4.4 combine these two sections- shorten and focus. Why do hedgehogs need more nests? Do they have parasites like fleas?

Thanks for valuable comments. We have combined these two sections as suggested. Interestingly, we did not observe any fleas, or other ectoparasites on the hedgehogs, and this was also the case for hedgehogs in Trondheim (further north in Norway) in 1994-95 (unpublished). However, nest switching might still be an anti-parasite strategy and we have discussed this briefly in the beginning of section 4.3 (l. 501-506)

585: Mortality- please delete this section -you have not studied mortality of hedgehogs- this is just speculation-

Thank you for the suggestion. We have removed this part.

586-588. Conclusion- Do only conclude on what you have data to conclude. Focus on home range and nest material. Please no speculations of general management

Thanks again for valuable comments, we have rewritten the conclusion to accommodate this.

Round 2

Reviewer 1 Report

Comments and Suggestions for Authors

There is not any questions with the revised manuscript.

Author Response

Thank you your valuable feedback during this process.

Reviewer 2 Report

Comments and Suggestions for Authors

I think that the manus has improved considerably and had become very interesting and valuable regarding suggestions to improve the habitat of hedgehogs in urban areas.

I have only minor suggestions:

line 11 use italic for the latin name

line 407 delete:  "Our results are in accordance with this" - You have described this above. 

416 "rather quickly" does not ad any knowlege. Instead write "within xx days"

502 "why nest sharing happens so rare among" remove "so" or "why nest sharing is rarely observed among hedgehogs"

517 a small character has to be removed behind the reference

559 what about garden owners can they provide shelters with nest material and should they make more nest posibilities in their gardens to improve the habitat of hedgehogs. You may add something about this.

Author Response

Response to the comments of reviewer 2

I think that the manus has improved considerably and had become very interesting and valuable regarding suggestions to improve the habitat of hedgehogs in urban areas.

Thank you for your valuable feedback throughout this process and for spotting the minor errors still present in the manuscript. Below you will find what we have done with each of the suggestions.

I have only minor suggestions:

line 11 use italic for the latin name
This has been changed.

line 407 delete:  "Our results are in accordance with this" - You have described this above. 
This has been deleted.

416 "rather quickly" does not ad any knowlege. Instead write "within xx days"
This has been changed.

502 "why nest sharing happens so rare among" remove "so" or "why nest sharing is rarely observed among hedgehogs"
This has been changed.

517 a small character has to be removed behind the reference
This has been deleted.

559 what about garden owners can they provide shelters with nest material and should they make more nest posibilities in their gardens to improve the habitat of hedgehogs. You may add something about this.
Thank you for this suggestion. We have now added the following sentence to the end of the conclusion: “. It would also be beneficial to encourage garden owners to contribute by providing shelters with ample natural nesting material in their gardens.”